# Nutritional status and effective verbal communication in Peruvian children: A secondary analysis of the 2019 Demographic and Health Survey

Akram Hernández-Vásquez[1], Nilthon Pisfil-Benites[2], Rodrigo Vargas-Fernández[3]*, Diego Azañedo[4]

**1** Centro de Excelencia en Investigaciones Económicas y Sociales, Vicerrectorado de Investigación, Universidad San Ignacio de Loyola, Lima, Peru, **2** Facultad de Economía y Planificación, Universidad Nacional Agraria de la Molina, Lima, Peru, **3** Universidad Científica del Sur, Lima, Peru, **4** Instituto de Investigación, Universidad Católica los Ángeles de Chimbote, Chimbote, Peru

* 100006379@ucientifica.edu.pe

**Data Availability Statement:** All relevant data are within the paper and its Supporting Information files.

## Abstract

### Background

To estimate the impact of stunting on the development of effective verbal communication (EVC) in children aged 24 to 36 months.

### Methods

We conducted a retrospective, quasi-experimental study design using data from 4452 children between 24 and 36 months of age available in the Early Childhood Development (ECD) section of the Demographic and Family Health Survey (ENDES) 2019 survey. Achievement of EVC was considered as the dependent variable. After propensity score matching analysis, a total 601 children were included in the exposed (children with stunting) and 3848 in the unexposed group (children without stunting). The *psmatch2* Stata software command was used to estimate the impact of stunting on EVC with a level of 5% for significance.

### Results

The EVC indicator was achieved by 57.4% of the children between 24 and 36 months of age evaluated, while the prevalence of stunting in this population was 14%. The evaluation of impact showed that the group exposed to stunting was 8% less likely to show EVC compared to the unexposed group (ATT -0.08; 95% CI -0.106 to -0.054; p<0.001).

### Conclusions

The presence of stunting was found to have a significant impact on the propensity to show EVC in Peruvian children between 24 and 36 months of age. Strengthening of strategies for

**Funding:** The authors received no specific funding for this work.

**Competing interests:** The authors have declared that no competing interests exist.

reducing malnutrition in vulnerable areas, as well as those directly affecting EVC is a priority for diminishing gaps in the achievement of this indicator in our country.

## Introduction

Child malnutrition produces serious metabolic and structural consequences for the individual and has a high social and economic impact [1], mainly in low- and middle-income countries, in which at least 50% of the mortality among children under five years of age is caused by malnutrition [2]. The chronicity of this condition affects a large proportion of children under five throughout the world, with an estimated 23% (144 million) of people in this age group presenting stunting in 2019 [3]. In 2011, low- and middle-income countries reported the highest proportions of malnutrition with 28% of stunting in children, in comparison to the 7.2% reported in high-income countries [4]. Also in 2017, Peru reached 12.9% of stunting in children under five years of age, which is under the average prevalence of 25% for this outcome in developing countries the same year [5]. This becomes relevant because the Early Childhood development (ECD) (perceptual, motor, cognitive, language, socio-emotional and self-regulation skills) is highly dependent on an active and healthy life, which cannot be well achieved in the presence of malnutrition [6].

ECD is target 4.2 of Goal 4 of the Sustainable Development Goals, which ensure that all children must achieve successful early childhood development in developing countries [7]. However, in these countries, nearly 150 million children under five years of age are at risk of failing to reach their potential development due to malnutrition and extreme poverty [8]. ECD is essential to achieve adequate learning, occupational productivity, physical and psychological health, and social well-being [9]. In Peru, the Ministry of Development and Social Inclusion, implemented a program aimed at improving ECD. This program included seven areas of development corresponding to a healthy birth, secure attachment, adequate nutritional status, effective verbal communication, walking alone, regulation of emotions and behaviors and symbolic function [8, 10].

Effective verbal communication (EVC) begins to develop at an early age, through crying or other forms of rudimentary communication as an immediate mechanism of interrelationship between children and parents. This communication influences adolescent development in the academic stage and work productivity as well as the development of emotions in adult life [11]. To achieve adequate EVC, the nutrition of a child from the prenatal period through the first 24 months of life must be adequate because the risk of stunting during this period in developing countries is high, leading to consequences in cognitive development [11]. Stunting also creates learning difficulties in the first one thousand days of life and has an impact on EVC in the long term [12, 13]. Therefore, in order to achieve the proper development of EVC, adequate nutrition and eating habits of the child must be ensured (during the first months of life) and those of the mother must be improved. Moreover, secure attachment of children and their parents is necessary as well as stimulation of motor developmental and gesticulation [14–16].

In Peru, there are no population-based studies on the impact of nutritional status on the development of EVC during childhood. However, the National Institute of Statistics and Informatics (INEI in Spanish) collects nutritional information on children under five years of age through the Demographic and Family Health Survey (ENDES in Spanish). During a recent validation of the ECD instrument the ENDES also included information on seven components of ECD including EVC [17]. Therefore, the objective of the present study was to estimate the effect of stunting in children aged 24 to 36 months on the development of EVC.

## Materials and methods

### Design and study population

The ENDES is a survey representative of the Peruvian population. In a secondary analysis of the ENDES 2019, the sample was characterized by being bi-stage, probabilistic of a balanced, stratified and independent type, at the departmental level and by urban and rural areas. The sample size of ENDES 2019 was 36,760 homes, 9320 of which were in urban areas and 12,660 were from rural areas.

The study had a retrospective, quasi-experimental design and included 4452 children between 24 and 36 months of age. The children were analyzed in the ECD section of the ENDES 2019, which evaluates different aspects of child development including EVC. To estimate the impact of stunting on EVC the sample was divided into exposed (children with stunting) and unexposed children (without stunting).

### Variables

The World Health Organization (WHO) classification (defined by variations of more than 2 standard deviations [SD] of the height for age below the 2006 WHO Child Growth Standards median) was used for the exposure variable "stunting" (HW70) [18] and EVC in children aged between 24 to 36 months (R4). The latter variable, measures EVC behaviors at the comprehensive and expressive level for each evolutionary stage of children aged 24 to 36 months according to a psychological instrument (Battelle Development Instrument) previously validated by experts who agreed on the questions to be included by the calculation of an Aiken indicator V greater than 0.7 (85.2%). This instrument was previously used in a pilot test in four Peruvian cities to determine adequate functioning and is based on questions that are performed and answered by the mothers [19].

The study covariates (codes in parentheses) included: age of the child in months (QI478), age of the mother (V012), years of education of the mother (V133), hemoglobin level of children (HW53), sex of the child (B4), total householders (HV009), health insurance coverage (V481), total children ever born (V201) and cesarean delivery (M17). In addition, the following variables: were constructed: "quintile" of the wealth index (V190) categorized into five levels (poorest, poor, middle, rich, richest), "prenatal control" according to prenatal visits (inadequate [less than 8] and suitable [8 or more]) during pregnancy (M14), "altitude" (m. a. s. l.) of the place of residence (HV040), "area" classifying the type of place of residence as urban or rural (V025), "ethnicity" (V131) considering ethnic origin (native and non-native), "natural region" including the natural region of origin (coast, highlands and jungle), "immediate lactation" considered as whether the infant was breastfed within the first hour after birth or not (V426), and "breastfeeding in months" according to the number of months the child was breastfed (M5). The covariates were selected according to revision of the literature available [20–28].

### Statistical analysis

The model proposed by Roy-Rubin [29, 30], was followed, defined as an exposure variable to stunting ($D_i$) and as an EVC outcome variable in children between 24 and 36 months ($Y_i$). That is, $Y_i(1)$ is the result variable if the individual $i$ was exposed and $Y_i(0)$ is the result variable if the individual $i$ was not exposed. Thus, the effect of exposure ($\tau_i$) for an individual $i$ can be calculated as:

$$\tau_i = Y_i(1) - Y_i(0)$$

The analysis focused on determining the average impact of exposure (Average Treatment Effect [ATE]) and the average treatment effect on the exposed population (Average Treatment on the Treated [ATT]).

$$\tau_{ATE} = E[Y_i(1) - Y_i(0)]$$

$$\tau_{ATT} = E[Y_i(1)D_i = 1] - E[Y_i(0)|D_i = 1]$$

Where: $E[.|D]$ corresponds to an expectation operator.

In this case, the average impact of exposure in exposed subjects ($\tau_{ATT}$) is the expected value of the difference in the result variable in the group of individuals exposed when there is the presence of exposure $E[Y_i(1)|D_i = 1]$ and the expected value of the result variable in the exposure group in the absence of exposure, known as contrafactual $E[Y_i(0)|D_i = 1]$. That is, the estimated impact is the difference in the average of the result variable of the exposure group and the control group.

The estimated impact is composed of the effect of exposure and selection bias. Formally, it is assumed that the selection bias is due only to differences in observable characteristics (conditional independence condition (IC)). What this assumption makes is to ensure that the selection bias is equal to zero, generating an unbiased estimate of the true effect of exposure. This is achieved by finding an individual exposed to stunting with the same observable characteristics in the control group.

An individual is an adequate control of an individual exposed to stunting, if both individuals are similarly likely to be exposed to stunting. This implies that the propensity score matching (PSM) method proposed by Rosenbaum Rubin (1983) can only be calculated in the Common Support Region (CS) to ensure that the exposure and control groups are very similar.

The PSM estimator is the average difference of the result variables of the individuals in the exposure group and the control group in the common support with an appropriate probability of exposure participation.

**Propensity and match estimation.** A logit model estimated the exposure probabilities for the covariates, i.e. the probability of participation with the observable characteristics of individuals. Having the probability of exposure and with the intention to compare the exposure impact results, a matching procedure was implemented with the kernel estimator that matches each respondent in the exposure group with a weighted average of all respondents in the control group. In addition, a matching procedure was estimated with the nearest neighbor, without replacement, using the estimated exposure probabilities, implementing a 1:1 correspondence structure. To improve the quality of the matching the condition that the maximum absolute difference between the exposure probabilities of matched respondents was not greater than 0.1 was imposed. Finally, an IPWRA matching procedure was implemented using weighted regression coefficients to calculate the impact of exposure, where weights are the estimated inverse probabilities of exposure.

**Post-match balancing diagnosis.** To ensure the balance of coincidence matching in the probability of exposure between the specified covariate distributions of the two exposure groups, two assumptions were required for correct estimation of the impact of exposure: a) the difference in the average of the exposure variable of the exposure and control group is composed of the effect of stunting and the selection bias, and b) any combination of characteristics observed in the exposure group also exists in the control group, ensuring that the exposure and control groups are very similar or are within a common support. Compliance with these assumptions is assumed for the estimation of the PSM.

Post-matching, the covariate distributions between the exposure groups in the matching sample were evaluated using the standardized absolute mean differences (bias) and variance ratios. After correspondence, standardized differences in the means of the covariates are expected to be close to zero, and variance ratios are expected to be close to one, if there is an adequate balance of covariates.

**Estimating the impact of stunting on outcomes.** The impact of stunting on children between 24 and 36 months of age was estimated by comparing the results between those exposed and unexposed in a subsample of individuals with similar characteristics. To do this, the *psmatch2* command was used. In the case of the PSM estimate per Kernel and the PSM estimate per nearest neighbor, the *teffects ipwra* command of Stata was also used for weighted effect estimates with adjusted regression, considering that all impact estimates were evaluated at the 5% significance level.

**Sensitivity analysis.** A sensitivity analysis was performed on the effect estimates using the limit approach of Rosenbaum. This approach explains that if there were the influence of unmeasured covariates, then a match between the two groups of covariates would not have the same probability of exposure allocation, and their probability ratio ($\Gamma$) would be different from one. By increasing the value of $\Gamma$ to identify the probabilities of exposure allocation ($\Gamma>1$), the degree of influence that an unmeasured covariate must have on the impact allocation and the validity of a study can be observed. Sensitivity analyses were performed for different values of $\Gamma>1$, in increments of 0.05, to determine the extent to which the $\Gamma$ value remained at a significance level of 0.05. The inferences are then considered sensitive to bias caused by non-measured covariates if the $\Gamma$ values are closer to 1 by altering the results (>0.05), and are considered relatively robust if values greater than $\Gamma$ are required to obtain results that affect inference. The *mhbounds* command of Stata was used for this sensitivity analysis [29].

## Ethical considerations

The study did not require the approval of an ethics committee because it is an analysis of secondary data that is in the public domain and does not allow the identification of the participants evaluated.

## Results

Table 1 shows the characteristics of the 4452 children included, of which 604 (13.6%) were exposed to stunting (exposure group) and 3848 (86.4%) were not exposed (control group). The mean current age of the mothers was 30 years, while the mean lactation period of the children was 19 months, and the mean hemoglobin level of the children was 12.2 g/dl. In addition, the highest proportion in the wealth index was the poor (28.3%) and very poor quintiles (26.1%); 27.8% were from rural areas, 42.2% were from the geographical area of the Coast; 82.6% had health insurance; and 57.4% of the children evaluated reached the EVC indicator.

Table 2 shows the distribution and comparison of covariates between the exposed and unexposed groups in matched and unmatched samples, of which four covariates had biases greater than +/- 0.75 after matching, while 10 covariates had differences +/- 0.25 in lower matching biases. Overall, bias of the model decreased from 31.1 to 2.0 after matching. Among exposed children, the altitude of the conglomerate, hemoglobin levels, live births and ethnicity were positively associated with exposure (0.01–0.03), while the current age of the mother, months of lactation, higher wealth quintile and residents in the jungle region (0.80–0.98) were less likely to be exposed.

As a result of the matching technique, all children found their unexposed counterpart, and the coincidence eliminated three observations. After matching, all covariates showed

**Table 1. Characteristics of Peruvian mothers and children aged 24 to 36 months (n = 4452).**

| Characteristics | n = 4452 |
|---|---|
| **Numeric variables** | |
| Current age in months, mean (SD) | 30.0 (3.7) |
| Current age of mother in years, mean (SD) | 30.2 (6.8) |
| Mother years education, mean (SD) | 10.5 (3.8) |
| Altitude of the place of residence in meters, mean (SD) | 1269.0 (1420.1) |
| Months of breastfeeding, mean (SD) | 19.1 (8.0) |
| Household size, mean (SD) | 5.1 (2.0) |
| Hemoglobin level of children, mean (SD) | 12.2 (1.3) |
| Children ever born, mean (SD) | 2.4 (1.5) |
| **Categorical variables** | |
| Q0 (richest), n (%) | 484 (10.9) |
| Q1, n (%) | 662 (14.9) |
| Q2, n (%) | 883 (19.8) |
| Q3, n (%) | 1260 (28.3) |
| Q4 (poorest), n (%) | 1163 (26.1) |
| Prenatal control*, n (%) | 2984 (67.0) |
| Male children, n (%) | 2256 (50.7) |
| Rural residence, n (%) | 1238 (27.8) |
| Coast, n (%) | 1878 (42.2) |
| Highlands, n (%) | 1466 (32.9) |
| Jungle, n (%) | 1108 (24.9) |
| Immediate lactation, n (%) | 2342 (52.6) |
| Native ethnicity, n (%) | 390 (8.8) |
| Health insurance | 3679 (82.6) |
| Cesarean delivery, n (%) | 1428 (32.1) |

Q4: poorest, Q3: poor, Q2: middle, Q1: rich, Q0: richest; SD: standard deviation

* 8 or more.

standardized means close to zero. The propensity score was estimated for each of the observations, and the balance was diagnosed, verifying the quality of the matching and imposing the common support. Fig 1 shows the assessment of the assumption of overlap.

## Estimation of exposure impact

The estimate of the impact of stunting is presented in Table 3. By way of comparison, the results of three different PSM estimators are presented. The estimated PSM Kernel shows the impact of stunting on children between 24 and 36 months of age, which indicates that the group exposed to stunting were 8% less likely to show EVC compared to the unexposed group, with a 95% confidence interval of -0.106 to -0.054 (95% CI). The nearest per-neighbor estimator (no replacement) and weighting with adjusted regression (IPWRA) showed similar results of 9% (95% CI: -0.144 to -0.034) and 6.3% (95% CI: -0.119 to -0.005), respectively. After matching, the covariates in the two groups the mean standardized biases decreased from 31.1 to 2.0 (Fig 2), of which 15 observed variables were in the range of (-0.05, +0.05) after matching, and 5 observed variables showed a higher matching to the aforementioned range. The matching bias in all covariates was less than (-0.98, +0.81).

**Table 2. Comparison of background covariates between exposure groups in the unmatched and matched samples of Peruvian mothers and children aged 24 to 36 months.**

| | | Unmatched | | | | Matched | | |
| | Exposure | Control | Difference | | Exposure | Control | Difference | |
| Characteristics | (n = 604) | (n = 3848) | t-stat | P-value | (n = 601) | (n = 3848) | t-stat | P-value |
|---|---|---|---|---|---|---|---|---|
| **Numeric variables** | | | | | | | | |
| Current age in months, mean (SD) | 30.018 | 30.026 | -0.05 | 0.962 | 30.005 | 30.059 | -0.25 | 0.800 |
| Current age of mothers in years, mean (SD) | 29.344 | 30.294 | -3.19 | 0.001 | 29.313 | 29.658 | -0.84 | 0.399 |
| Mother years education, mean (SD) | 8.066 | 10.914 | -17.64 | <0.001 | 8.102 | 8.252 | -0.66 | 0.509 |
| Altitude of the place of residence in meters, mean (SD) | 1950.400 | 1162.000 | 12.92 | <0.001 | 1944.500 | 1946.100 | -0.02 | 0.986 |
| Months of breastfeeding, mean (SD) | 18.745 | 19.167 | -1.21 | 0.227 | 18.772 | 19.189 | -0.98 | 0.325 |
| Household size, mean (SD) | 5.545 | 5.026 | 6.03 | <0.001 | 5.531 | 5.497 | 0.27 | 0.791 |
| Hemoglobin level of children, mean (SD) | 12.462 | 12.163 | 5.08 | <0.001 | 12.463 | 12.457 | 0.07 | 0.943 |
| Children ever born, mean (SD) | 3.073 | 2.350 | 11.12 | <0.001 | 3.052 | 3.052 | -0.01 | 0.994 |
| **Categorical variables** | | | | | | | | |
| Q0 (richest), n (%) | 0.028 | 0.121 | -6.88 | <0.001 | 0.028 | 0.038 | -0.91 | 0.364 |
| Q1, n (%) | 0.053 | 0.164 | -7.15 | <0.001 | 0.053 | 0.058 | -0.34 | 0.730 |
| Q2, n (%) | 0.083 | 0.216 | -7.71 | <0.001 | 0.083 | 0.088 | -0.29 | 0.771 |
| Q3, n (%) | 0.253 | 0.288 | -1.74 | 0.081 | 0.255 | 0.243 | 0.46 | 0.648 |
| Q4 (poorest), n (%) | Ref. | - | - | - | Ref. | - | - | - |
| Prenatal control*, n (%) | 0.556 | 0.688 | -6.44 | <0.001 | 0.559 | 0.572 | -0.45 | 0.650 |
| Male children, n (%) | 0.531 | 0.503 | 1.31 | 0.191 | 0.531 | 0.534 | -0.09 | 0.925 |
| Rural residence, n (%) | 0.565 | 0.233 | 17.47 | <0.001 | 0.562 | 0.550 | 0.44 | 0.660 |
| Coast, n (%) | Ref. | - | - | - | Ref. | - | - | - |
| Highlands, n (%) | 0.541 | 0.296 | 12.12 | <0.001 | 0.539 | 0.543 | -0.13 | 0.896 |
| Jungle, n (%) | 0.280 | 0.244 | 1.89 | 0.059 | 0.281 | 0.260 | 0.81 | 0.418 |
| Immediate lactation, n (%) | 0.652 | 0.506 | 6.72 | <0.001 | 0.652 | 0.652 | 0 | 0.998 |
| Native ethnicity, n (%) | 0.219 | 0.067 | 12.45 | <0.001 | 0.216 | 0.216 | 0.03 | 0.979 |
| Health insurance, n (%) | 0.894 | 0.816 | 4.73 | <0.001 | 0.894 | 0.891 | 0.16 | 0.875 |
| Cesarean delivery, n (%) | 0.185 | 0.342 | -7.71 | <0.001 | 0.186 | 0.196 | -0.43 | 0.668 |
| Mean standardized bias | | 31.1 | | | | 2.0 | | |
| Pseudo R2 (logit) | | 0.163 | | | | 0.002 | | |

Q4: poorest, Q3: poor, Q2: middle, Q1: rich, Q0: richest; SD: standard deviation.

* 8 or more.

## Sensitivity analysis

Table 4 shows a sensitivity analysis of Rosenbaum limits for estimating the impact of stunting on EVC in children aged 24 to 36 months. Under the assumption that there is no influence of non-observable (non-measured) variables, the rate of possibility ($\Gamma$) was 1 and the estimate of the effect was significant. If there were factors that had not been considered in the modeling that affect the response variable, stunting in children aged 24 to 36 months was less likely to have EVC, then the current estimate would have minimized the effects of exposure. Under the assumption of underestimation, a seamless variable would have to influence the odds exposure ratio to differ between the two exposure groups by a factor of more than 1.2 so that the estimate is not significant at the 5% level. However, assuming an overestimation, the estimate remained significant at all levels considered to be $\Gamma$ suggesting that it was unlikely to be overestimated.

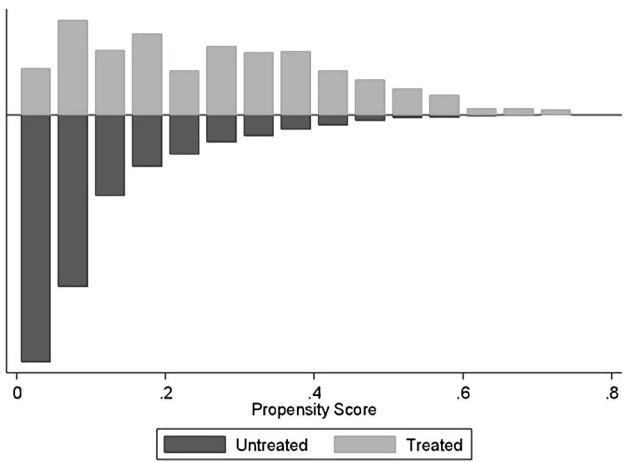

**Fig 1. Propensity score histogram.**

## Discussion

This study aimed to estimate the impact of stunting on the development of EVC in Peruvian children aged 24 to 36 months. After making PSM estimates to match groups of children exposed and not exposed to malnutrition, children exposed to stunting were found to be significantly (8%) less likely to show EVC, compared to children of the same age not exposed to malnutrition. These results show that good nutritional status is a key element for adequate ECD. Indeed, according to our analysis, in 2019 almost 14% of Peruvian children aged 24 to 36 months suffered from stunting. The presence of this condition could affect proper ECD in Peruvian children, having an impact on language skills and social interaction, which are important for adequate regulation of emotions and behaviors as well as the development of symbolic function (mental representation / mental scheme), and reading and writing skills.

To date, most studies have focused on assessing the impact of malnutrition on subrogated outcomes such as childhood anemia. Our study is one of the first to assess the impact of malnutrition on an objective outcome of the ECD indicator such as EVC, measured by an instrument validated in Peruvian children [19]. The results showed a negative impact of stunting on EVC, which in turn may have an impact on the development of other indicators of stunting. This finding can have long-term relevance, potentially in the school or university year and in social development in adults. Therefore, it is necessary to propose strategies that mitigate stunting as one of the factors influencing cognitive development with the aim of improving early childhood development and labor productivity in developing countries, like Peru.

**Table 3. Exposure effects of stunting on effective verbal communication in 601 Peruvian children aged 24 to 36 months.**

| Variable | Exposure | Control | ATT | t test | SE | Z | p value | 95% CI |
|---|---|---|---|---|---|---|---|---|
| Kernel | 0.489 | 0.569 | -0.080 | -3.31 | 0.024 | -6.00 | <0.001 | -0.106 to -0.054 |
| Noreplacement (0.1) | 0.489 | 0.579 | -0.09 | -3.07 | 0.029 | -3.20 | <0.001 | -0.144 to -0.034 |
| Ipwra | | | 0.063 | | 0.03 | -2.16 | <0.031 | -0.119 to -0.005 |

ATT: Average treatment on the treated.

SE: Standard error

CI: Confidence interval.

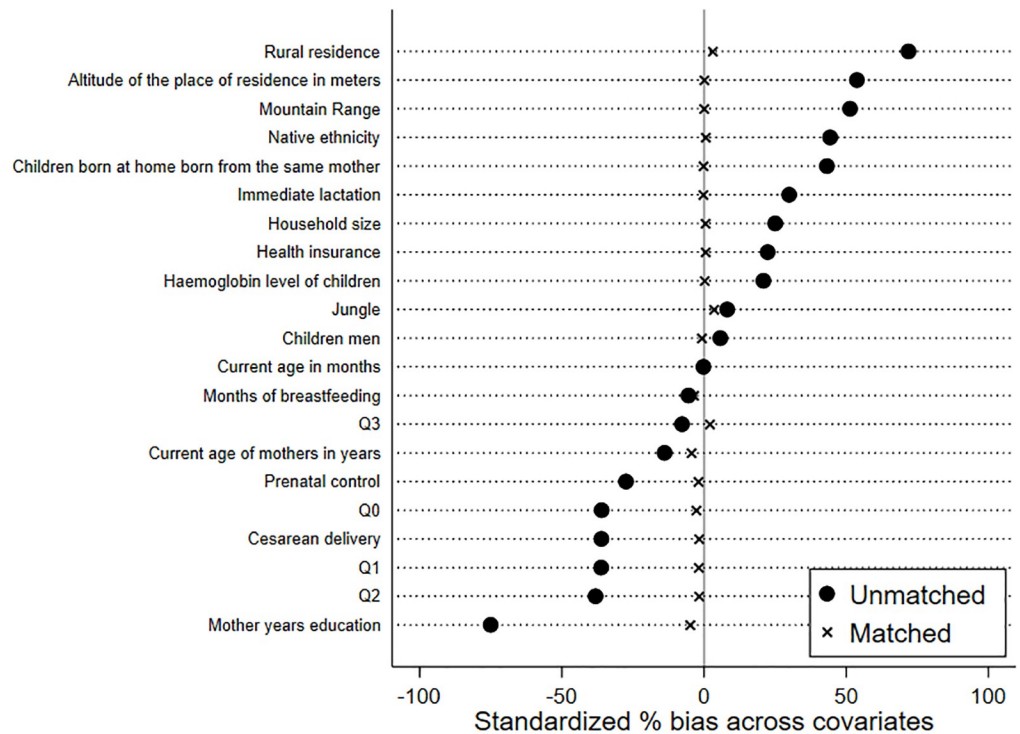

**Fig 2. Covariates balance before and after matching.**

It was also identified that 40% of Peruvian children between two and three years old would not develop adequate EVC, partly as a result of stunting. In this regard, it should be noted that the components of the ECD have intricate mechanisms that interact as the infant matures. For example, secure attachment of children and parents, and stimulation of motor development and gesticulation have been identified as being related to EVC [14–16]. Similarly, adequate nutritional development has a cross-over effect on the development of other indicators of ECD, one of the main being EVC, since it has been reported that stunting is associated with poor cognitive development, late school admission and decreased schooling, likely as a consequence of poor EVC development early in life [31]. In this sense, EVC is a problem that can only be effectively addressed by multidisciplinary and multisectoral approaches.

Similarly, a technical report from the INEI in 2018 reported a higher proportion of children aged 9 to 36 months who achieved EVC among residents of urban compared to rural areas (50.5% vs. 47.7%, respectively), children residing in the jungle vs. the coast and mountain areas (53% vs. 51.9% and 47%, respectively), children of mothers with higher vs. secondary and without education (53% vs. 49.3 and 45%, respectively), boys of the intermediate vs. lower economic quintile (52.3% vs. 46.2%, respectively) and girls vs. boys (54% vs. 45.6%, respectively) [32]. These results demonstrate the presence of gaps according to the sociodemographic characteristics of Peruvian children which must be taken into account by the government in order to target strategies to ensure that a higher proportion of children achieve EVC indicators and to reduce the inequalities in achieving this indicator. It has been mentioned that EVC can be influenced by different affective variables, such as secure attachment of children and their parents, and stimulation of motor development and gestures. However, our research included children ranging between 24 and 36 months of age, while the secure attachment variable was

**Table 4. Sensitivity analysis for estimating the impact of stunting on effective verbal communication in 601 Peruvian children aged 24 to 36 months.**

| Gamma Γ | p_mh+ | p_mh- |
|---|---|---|
| 1.000 | 0.000 | 0.001 |
| 1.050 | 0.000 | 0.005 |
| 1.100 | 0.000 | 0.014 |
| 1.150 | 0.000 | 0.035 |
| 1.200 | 0.000 | 0.073 |
| 1.250 | 0.000 | 0.133 |
| 1.300 | 0.000 | 0.218 |
| 1.350 | 0.000 | 0.323 |
| 1.400 | 0.000 | 0.440 |
| 1.450 | 0.000 | 0.489 |
| 1.500 | 0.000 | 0.377 |
| 1.550 | 0.000 | 0.277 |
| 1.600 | 0.000 | 0.195 |
| 1.650 | 0.000 | 0.131 |
| 1.700 | 0.000 | 0.085 |
| 1.750 | 0.000 | 0.053 |
| 1.800 | 0.000 | 0.032 |
| 1.850 | 0.000 | 0.018 |
| 1.900 | 0.000 | 0.010 |
| 1.950 | 0.000 | 0.006 |
| 2.000 | 0.000 | 0.003 |

Gamma Γ: Odds of differential assignment due to unobserved factors.

p_mh+: significance level (assumption: overestimation of exposure effect).

p_mh-: significance level (assumption: overestimation of exposure effect).

collected only in children under 12 months, making it impossible to assess how this variable could affect our results. Therefore, EVC is one of the seven components of ECD and its measurement is of interest between 9 and 36 months of age. Indeed, in order to facilitate the work of decision-makers, national studies and technical reports should report stunting according to age groups and not aggregated for all children under five years of age (60 months). This would enable focusing on nutritional and complementary strategies for the development of each of the components of the ECD according to specific age groups with potential vulnerability.

Stunting continues to be a problem in Peruvian children, with 14% of children between 24 and 36 months suffering from this condition in 2019. This prevalence is above the 9.6% of stunting reported in Latin America and the global prevalence of 13.5% in children under five years old in 2017 [33, 34]. In addition, it has been described that in the first half of 2019, 33.2% of Peruvian children aged between 24 to 36 months were at risk of stunting [35]. Although our country has shown a marked reduction in the prevalence of stunting from 30% in 2005 to 15% in 2015 [36], these results show that efforts are still needed to reduce the prevalence of this condition due to the absence of substantial reductions between 2015 and 2019, and the considerable number of children who remain at risk of stunting. Likewise, attention must be paid to the Peruvian Andean region, where according to information from 2018 the lowest proportion of children meet the EVC indicator [32], with a prevalence of stunting of up to 30.1% and a risk of stunting in children under five years old of 58.2% [31].

This study has limitations. The methodological limitation related to the PSM is based on the application of Common Support (CS) since it tends to eliminate observations in order to achieve a completely random experiment. In this study, the impact of stunting in children between 24 and 36 months was estimated, with the application of the PSM by Kernel and the estimate by nearest neighbor, and the estimate of weighted effects with adjusted regression was considered. In addition, a sensitivity analysis was applied to the effect estimates using the Rosenbaum limit approach. These estimates were made with the intention of assessing the impact of the exposure under a different methodological perspective. Likewise, few observations were removed compared to the more than 4000 observations included in the study. On the other hand, the use of the secondary database of ENDES for analysis may introduce information biases by the respondents when providing the requested information as these may have provided inaccurate or false information. In addition, this bias may be present due to erroneous introduction of data at the time of data registration by the interviewers. Moreover, since the sample weights of the survey were not included in our analyses, the results cannot be generalized to the ENDES target population. Despite these limitations, it should be noted that ENDES follows the standardized design and procedures of the Demographic and Health Survey (DHS) program, which includes training of interviewers to reduce the introduction of biases during information collection in order to ensure the quality of the data collected. Similarly, a PSM analysis was used, providing a better balance of covariate measures in groups of children exposed and unexposed to stunting with the aim of reducing the risk of confusion bias. This study is also one of the first national reports assessing the association between stunting and an objective outcome of the ECD, such as EVC, with representative data. However, it must be taken into account that due to the complexity of the measurement of ECD, its components, and causality networks, a deeper understanding of the contribution of stunting over different components of ECD is needed to establish stronger and more robust conclusions on this association.

## Conclusion

In conclusion, the presence of stunting has a significant negative impact on the propensity to develop EVC in Peruvian children between 24 to 36 months of age. Strategies for reducing malnutrition should be strengthened in population areas identified as vulnerable and should prioritize the components of ECD according to age groups for which their measurement is of interest to complement nutritional strategies with those that directly concern EVC. This would reduce the gaps in EVC among children in Peru with the aim of ensuring adequate ECD and equal opportunities for all Peruvian children.

## Supporting information

**S1 Table. Database.**
(XLS)

## Acknowledgments

The authors are grateful to Elaine Pringle Donna for reviewing the language and style.

## Author Contributions

**Conceptualization:** Akram Hernández-Vásquez.

**Data curation:** Akram Hernández-Vásquez.

**Formal analysis:** Akram Hernández-Vásquez, Nilthon Pisfil-Benites.

**Methodology:** Akram Hernández-Vásquez, Nilthon Pisfil-Benites.

**Supervision:** Akram Hernández-Vásquez.

**Validation:** Akram Hernández-Vásquez, Nilthon Pisfil-Benites, Rodrigo Vargas-Fernández, Diego Azañedo.

**Writing – original draft:** Akram Hernández-Vásquez, Nilthon Pisfil-Benites, Rodrigo Vargas-Fernández, Diego Azañedo.

**Writing – review & editing:** Akram Hernández-Vásquez, Nilthon Pisfil-Benites, Rodrigo Vargas-Fernández, Diego Azañedo.

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
