## [Decision Letter · Decision Letter 0]

7 Oct 2020

PONE-D-20-26209

Impact of nutritional status on effective verbal communication in Peruvian children. A nationwide study in Peru

PLOS ONE

Dear Dr. Vargas-Fernández,

Thank you for submitting your manuscript to PLOS ONE. After careful consideration, we feel that it has merit but does not fully meet PLOS ONE’s publication criteria as it currently stands. Therefore, we invite you to submit a revised version of the manuscript that addresses the points raised during the review process.

These are primarily edits to the text to increase clarity; they do not appear to require collection of any new data.

We look forward to receiving your revised manuscript.

Kind regards,

James West, PhD

Academic Editor

PLOS ONE

Journal Requirements:

Reviewers' comments:

Reviewer's Responses to Questions

**Comments to the Author**

1. Is the manuscript technically sound, and do the data support the conclusions?

Reviewer #1: Yes

Reviewer #2: Yes

2. Has the statistical analysis been performed appropriately and rigorously? 

Reviewer #1: Yes

Reviewer #2: Yes

3. Have the authors made all data underlying the findings in their manuscript fully available?

Reviewer #1: No

Reviewer #2: Yes

4. Is the manuscript presented in an intelligible fashion and written in standard English?

Reviewer #1: Yes

Reviewer #2: Yes

5. Review Comments to the Author

Reviewer #1: In Table 1 & 2:

The caption should be complete including the which study population and study site. The questions items should be described in the table and mean/SD should be reported separately. Some questions have been labelled Q1 Q2 that should be descried in this table.

In Table 3, the caption should include Effective Verbal Communication instead of its abbreviation including the study population, study site and sample size.

Table 4: Revise and complete the caption of the table, suggested “sensitivity analysis for estimating the impact of chronic 236 malnutrition on EVC in children aged 24 to 36 months”.

Study Limitation and its strengths should be mentioned under a spate heading after end of the research paper.

The study title should reflect that it is a secondary data analysis so accordingly study title should be revised.

Reviewer #2: Thank you for the opportunity to read this interesting manuscript that estimates the impact of chronic malnutrition on the development of verbal communication in toddler aged children.

Overall, I think it is a great and well-executed paper. I have a few comments.

Introduction.

I really enjoy reading about the Peruvian context and what is being done in Peru in term of supporting the ECD agenda. Here are some minor comments about things that were a little unclear:

1.

“Child malnutrition produces serious metabolic and structural consequences for the individual and has a high social and economic impact [1], mainly in low- and middle-income countries, where nearly half of the children under five years of age die due to malnutrition [2].”

This sentence could be improved to be more clear. I read this like you are saying that 50% of all children who are under 5 years in LMIC die due to malnutrition. I suppose that what you mean is that when 50% of the mortality among children <5 years in LMIC is caused by malnutrition?

2.

“This becomes relevant because children integral development at an early stage (perceptual, motor, cognitive, language, socio-emotional and self-regulation skills), is highly dependent of an active and healthy life, which cannot be well achieved in the presence of malnutrition [6].”

What do you mean by integral development?

3.

“Early childhood development (ECD) is one of the goals of the Sustainable Development Goals in developing countries [7].:

Can you be more specific as you which goal(s) you refer to? Or do you mean more generally embedded into the SDGs?

4.

“In Peru, the government, led by the Ministry of Development and Social Inclusion, implemented a program aimed at improving ECD. This program included seven areas of development corresponding to a healthy birth, safe attachment, adequate nutritional status, effective verbal communication, walking alone, regulation of emotions and behaviors and symbolic function [8,10].”

This is quite a minor comment, but to me this sounds like the Peruvian government is led by a ministry. What I believe you are trying to say that the ministry led this initiative.

Methods

This is a great use of publicly available data and I like for that you mention the specific variable names you have used for replicability although it seems a little unconventional.

5.

I would like a little more information about both the exposure variable and the outcome variable.

For “chronic malnutrition” you simply say that you used WHO’s classification, but what exactly does that mean? Malnutrition is defined as deficient intake of nutrients (macro and micro) but did you use an index of food intake or did you measure an anthropometric proxy like stunting, wasting, middle upper arm circumference? This is important to understand and more clarity is needed. Relatedly, how to you access chronicity? This may be imbedded in a variable like stunted growth which is believed to reflect nutrition over time.

For EVC, you state that it is “performed and answered” by mothers. Exactly what does “performed” mean in this regard? Did you test the child on certain things? How many items does the outcome integrate? Could these maybe be listed in an appendix?

6.

You include a lot of covariates in the analysis and I wonder if all were necessary. A stronger and more specific rationale would be helpful. Are some of these chosen because they may predict child growth status rather than language? E.g., cesarean delivery, prenatal control, "altitude", "immediate lactation"?

Statistical analysis

I really appreciate the level of detail provided in the section. I also appreciate the use of three different methods to compare results.

7.

Propensity and match estimation : what is the rationale for the 0.1 cut-off? It seems that are more conservative 0.05 or 0.01 is quite common.

8.

Page 19, line 248 : might you mean under-estimation?

Results

No comments

Discussion:

10.

You state that 40% of Peruvian children between two and three years old would not develop adequate

EVC, partly as a result of chronic malnutrition. If malnutrition only explains 8% then it does not seem to be the main source of developmental setbacks in communication?

Relatedly, please comment on the extent to which a 8% lower prevalence of EVC is substantial and problematic. This seems like a pretty small effect size. The importance of this effect would also be easier to grasp if you include more information about the outcome variable.

You mention some other variables like attachment and socioeconomic factors that may affect EVC, but these would be accounted for by the matching, correct? So there still seems to be an unexplained gap?

11.

I agree that the PSM approach is a great strength, but you may discuss some of the limitations of PSM methodologies. King and Nielsen (2016) proposed that PSM has serious limitations. Despite their analysis mostly refers to pair matching without replacement, they made interesting points on the fully blocked randomization approach. This is highly debatable, but still a good way to discuss PSM limitations. In particular given the current trends point toward the use of weighting, something used as a secondary approach.

6. PLOS authors have the option to publish the peer review history of their article (what does this mean?). If published, this will include your full peer review and any attached files.

Reviewer #1: **Yes: **Subhana Akber Khan

Reviewer #2: No

---

## [Author Response · Author response to Decision Letter 0]

22 Dec 2020

Dr. James West

Academic Editor

PLOS ONE

Title: Impact of nutritional status on effective verbal communication in Peruvian children. A nationwide study in Peru

New title: Nutritional status and effective verbal communication in Peruvian children. A secondary analysis of the 2019 Demographic and Health Survey

Dear Editor and Reviewers,

Thank you for your valuable suggestions and remarks. After incorporating your feedback our manuscript has greatly improved. Please find below the point-by-point responses to your comments. We hope that the present version of the manuscript is now acceptable for publication, and we look forward to your feedback.

Comments of reviewers:

Introduction.

I really enjoy reading about the Peruvian context and what is being done in Peru in term of supporting the ECD agenda. Here are some minor comments about things that were a little unclear: 

Comment 1:

“Child malnutrition produces serious metabolic and structural consequences for the individual and has a high social and economic impact [1], mainly in low- and middle-income countries, where nearly half of the children under five years of age die due to malnutrition [2].”

This sentence could be improved to be more clear. I read this like you are saying that 50% of all children who are under 5 years in LMIC die due to malnutrition. I suppose that what you mean is that when 50% of the mortality among children <5 years in LMIC is caused by malnutrition? 

Response: Thank you for the observation. Your assumption is correct, for better understanding we have rewritten the sentence.

Now it reads: "Child malnutrition produces serious metabolic and structural consequences for the individual and has a high social and economic impact [1], mainly in low- and middle-income countries, in which at least 50% of the mortality among children under five years of age is caused by malnutrition [2].”

Comment 2:

“This becomes relevant because children integral development at an early stage (perceptual, motor, cognitive, language, socio-emotional and self-regulation skills), is highly dependent of an active and healthy life, which cannot be well achieved in the presence of malnutrition [6].”

What do you mean by integral development?

Response: Thank you for the observation. By integral development we meant Early childhood development, we have corrected this term so as not to introduce confusion. Early childhood development (ECD) refers to the maturation processes. These mainly develop during the first two years of life, during which the greatest brain development is observed, resulting in the progression and achievement of a group of skills that children acquire as their age advances (perceptual, motor, cognitive, language, socio-emotional and self-regulation skills) [1,2], and requires an optimal home environment and adequate nutrition [3].

1. Black MM, Walker SP, Fernald LCH, et al. Early childhood development coming of age: science through the life course. Lancet. 2017;389(10064):77-90. doi:10.1016/S0140-6736(16)31389-7.

2. Bundy DAP, de Silva N, Horton S, et al. Investment in child and adolescent health and development: key messages from Disease Control Priorities, 3rd Edition. Lancet. 2017;391(10121):687-699. doi: 10.1016/S0140-6736(17)32417-0.

3. Alderman H, Fernald L. The Nexus Between Nutrition and Early Childhood Development. Annual Review of Nutrition. 2017;37(1):447–476. doi:10.1146/annurev-nutr-071816-064627.

Change made: The sentence now reads “This becomes relevant because the Early Childhood development (ECD) (perceptual, motor, cognitive, language, socio-emotional and self-regulation skills) is highly dependent on an active and healthy life, which cannot be well achieved in the presence of malnutrition [6].”

Comment 3:

“Early childhood development (ECD) is one of the goals of the Sustainable Development Goals in developing countries [7].: 

Can you be more specific as you which goal(s) you refer to? Or do you mean more generally embedded into the SDGs?

Response: Thank you for the observation. According to what was raised by the reviewer, the information regarding the SDGs has been drafted in more detail. 

Now it reads: “Early childhood development (ECD) is target 4.2 of Goal 4 of the Sustainable Development Goals, which ensure that all children must achieve successful early childhood development in developing countries [7].”

Comment 4:

“In Peru, the government, led by the Ministry of Development and Social Inclusion, implemented a program aimed at improving ECD. This program included seven areas of development corresponding to a healthy birth, safe attachment, adequate nutritional status, effective verbal communication, walking alone, regulation of emotions and behaviors and symbolic function [8,10].”

This is quite a minor comment, but to me this sounds like the Peruvian government is led by a ministry. What I believe you are trying to say that the ministry led this initiative. 

Response: Thank you for the observation. We agree with what was established by the reviewer, and the corresponding change has been made in the text for better understanding.

Now it reads: “In Peru, the Ministry of Development and Social Inclusion implemented a program aimed at improving ECD. This program included seven areas of development corresponding to a healthy birth, secure attachment, adequate nutritional status, effective verbal communication, walking alone, regulation of emotions and behaviors and symbolic function [8,10].”

Methods

This is a great use of publicly available data and I like for that you mention the specific variable names you have used for replicability although it seems a little unconventional.

Comment 5:

I would like a little more information about both the exposure variable and the outcome variable.

For “chronic malnutrition” you simply say that you used WHO’s classification, but what exactly does that mean? Malnutrition is defined as deficient intake of nutrients (macro and micro) but did you use an index of food intake or did you measure an anthropometric proxy like stunting, wasting, middle upper arm circumference? This is important to understand and more clarity is needed. Relatedly, how to you access chronicity? This may be imbedded in a variable like stunted growth which is believed to reflect nutrition over time. 

Response: Thank you for the observation. For the estimation of nutritional status, the anthropometric indicator of height for age of the World Health Organization (WHO) was considered, which reflects the growth achieved in height for the age of the child at a certain time. Chronic malnutrition was defined by variations of more than 2 standard deviations (SD) of the height for age below the WHO Child Growth Standards median.

Changes made: none.

Comment 6:

For EVC, you state that it is “performed and answered” by mothers. Exactly what does “performed” mean in this regard? Did you test the child on certain things? How many items does the outcome integrate? Could these maybe be listed in an appendix?

Response: As for the psychological instrument (Battelle Development Instrument) applied to identify Effective Verbal Communication (EVC) in girls and boys under 36 months, these questions were answered exclusively by the mothers. This means no tests were conducted directly in the children. However, the calculation of EVC was not part of the objective of the study and the results of whether EVC was achieved or not were taken directly from the database. Likewise, the survey is made up of 21 items that evaluate the impact of nutritional status on EVC in boys and girls in Peru, with detailed information on this being available on the website (http://iinei.inei.gob.pe/microdatos/). Given this, the wording of the article has been modified so that it is understood that the mothers of the boys and girls only answered the questions, but did not ask. On the other hand, in the study, children were only evaluated for the collection of the variables “hemoglobin level” and “sex of the boy or girl”, and for the other covariates the mother of each child was consulted. 

Change made: none

Comment 7: 

You include a lot of covariates in the analysis and I wonder if all were necessary. A stronger and more specific rationale would be helpful. Are some of these chosen because they may predict child growth status rather than language? E.g., cesarean delivery, prenatal control, "altitude", "immediate lactation"?

Response: Thank you for the observation. According to the Technical document of systematization of evidence to achieve early childhood development and previous studies, there are different variables that compose and predict child growth status and effective verbal communication, and among others, the variables for mother and child are sex of the child [1] and age of the mother [2], ethnicity [3], birth order, education and poverty of the mother [1], immediate lactation [4], prenatal control, cesarean delivery [5] and health insurance coverage [6]; while the variables for the household are area of residence [2], natural regions of origin [7], the altitude of the place of residence [8] and quintile of wealth index by the characteristics of the home [9]. In the description of variables in the Methods section, each of the studies that refer to their importance with the dependent and independent variables have been cited.

1. Sobrino M, Gutiérrez C, Alarcón J, et al. Birth interval and stunting in children under five years of age in Peru (1996-2014). Child Care Health Dev. 2017;43(1):97-103. doi: 10.1111/cch.12420. 

2. Crookston BT, Dearden KA, Alder SC, et al. Impact of early and concurrent stunting on cognition. Matern Child Nutr. 2011;7(4):397-409. doi:10.1111/j.1740-8709.2010.00255.x

3. Mena-Meléndez L. Ethnoracial child health inequalities in Latin America: Multilevel evidence from Bolivia, Colombia, Guatemala, and Peru. SSM Popul Health. 2020 Oct 3;12:100673. doi: 10.1016/j.ssmph.2020.100673. 

4. Marquis GS, Habicht JP. Breastfeeding and Stunting Among Toddlers in Peru. In: Koletzko B., Michaelsen K.F., Hernell O. (eds) Short and Long Term Effects of Breast Feeding on Child Health. Advances in Experimental Medicine and Biology. Springer, Boston, MA; 2002.

5. Saaka M, Hammond AY. Caesarean Section Delivery and Risk of Poor Childhood Growth. J Nutr Metab. 2020;2020:6432754. doi:10.1155/2020/6432754

6. Nshakira-Rukundo E, Mussa EC, Gerber N, et al. Impact of voluntary community-based health insurance on child stunting: Evidence from rural Uganda. 2020;245. doi: 10.1016/j.socscimed.2019.112738

7. Urke HB, Bull T, Mittelmark MB. Socioeconomic status and chronic child malnutrition: Wealth and maternal education matter more in the Peruvian Andes than nationally. Nutr Res. 2011;31(10):741-7. doi: 10.1016/j.nutres.2011.09.007. PMID: 22074798.

8. Loli S, Carcamo CP. Rotavirus vaccination and stunting: Secondary Data Analysis from the Peruvian Demographic and Health Survey. Vaccine. 2020 Nov 25;38(50):8010-8015. doi: 10.1016/j.vaccine.2020.10.044. 

9. Dearden KA, Schott W, Crookston BT, et al. Children with access to improved sanitation but not improved water are at lower risk of stunting compared to children without access: a cohort study in Ethiopia, India, Peru, and Vietnam. BMC Public Health. 2017;17(1):110. doi: 10.1186/s12889-017-4033-1. 

Comment 8:

Statistical analysis 

I really appreciate the level of detail provided in the section. I also appreciate the use of three different methods to compare results. 

Comment 9:

Propensity and match estimation : what is the rationale for the 0.1 cut-off? It seems that are more conservative 0.05 or 0.01 is quite common. 

Response: Thank you for the observation. The estimation of a propensity score is defined as the probability of a subject to receive a specific treatment (exposure) conditioned on the observed covariates [1-3], and matching is a method to reduce biases in the groups (control and exposure). Therefore, to select a treated subject and the closest untreated subject that is within the specified caliper, this is chosen as a coincidence in such a way that the literature proposes calibrators of 0.6, 0.2, 0.1, 0.001 and 0.005. The rationale for these choices was provided by Cochran and Rubin, who demonstrated that matching of a normally distributed confounding variable using calibrators removes 90% to 99% of bias due to confounding variables [4]. Other authors in the medical literature have matched the propensity score using width calipers of 0.005 [5-12], 0.01 [8-12], 0.02 [13], 0.03 [14] and 0.1 [15] in the propensity score scale. It is suggested that a standardized difference greater than 0.1 represents a significant imbalance difference in the given covariates between the groups.

1. Austin PC, Mamdani MM, Stukel TA. The use of the propensity score for estimating treatment effects: administrative versus clinical data. Statistics in Medicine. 2005;24(10):1563-1578. doi: 10.1002/sim.2053.

2. Rosenbaum P, Rubin D. The Central Role of the Propensity Score in Observational Studies for Causal Effects. Biometrika, 1983.

3. Rosenbaum P, Rubin D. Reducing Bias in Observational Studies Using Subclassification on the Propensity Score. Journal of the American Statistical Association, 1984.

4. Cochran WG, Rubin D. Controlling Bias in Observational Studies: A Review. Sankhyā: The Indian Journal of Statistics, Series A (1961-2002). 2020. 

5. Christakis NA, Iwashyna TJ. The health impact of health care on families: a matched cohort study of hospice use by decedents and mortality outcomes in surviving, widowed spouses. Soc Sci Med. 2003;57(3):465-75. doi: 10.1016/s0277-9536(02)00370-2.

6. Cole JA, Loughlin JE, Ajene AN, et al. The effect of zanamivir treatment on influenza complications: a retrospective cohort study. Clin Ther. 2002;24(11):1824-39. doi: 10.1016/s0149-2918(02)80082-0.

7. Iwashyna TJ, Lamont EB. Effectiveness of adjuvant fluorouracil in clinical practice: a population-based cohort study of elderly patients with stage III colon cancer. J Clin Oncol. 2002;20(19):3992-8. doi: 10.1200/JCO.2002.03.083. 

8. Hall JA, Summers KH, Obenchain RL. Cost and utilization comparisons among propensity score-matched insulin lispro and regular insulin users. J Manag Care Pharm. 2003;9(3):263-8. doi: 10.18553/jmcp.2003.9.3.263. 

9. Magee MJ, Coombs LP, Peterson ED, et al. Patient selection and current practice strategy for off-pump coronary artery bypass surgery. Circulation. 2003;108 Suppl 1:II9-14. doi: 10.1161/01.cir.0000089187.51855.77. 

10. Seeger JD, Walker AM, Williams PL, et al. A propensity score-matched cohort study of the effect of statins, mainly fluvastatin, on the occurrence of acute myocardial infarction. Am J Cardiol. 2003;92(12):1447-51. doi: 10.1016/j.amjcard.2003.08.057. 

11. Ferguson TB, Coombs LP, Peterson ED. Internal thoracic artery grafting in the elderly patient undergoing coronary artery bypass grafting: Room for process improvement?. The Journal of Thoracic and Cardiovascular Surgery. 2002;123(5):869-880. doi: 10.1067/mtc.2002.121679

12. Weiss JP, Saynina O, McDonald KM, et al. Effectiveness and cost-effectiveness of implantable cardioverter defibrillators in the treatment of ventricular arrhythmias among medicare beneficiaries. The American Journal of Medicine. 2002;112(7):519-527. doi: 10.1016/S0002-9343(02)01078-1.

13. Murray PK, Singer M, Dawson NV, et al. Outcomes of Rehabilitation Services for Nursing Home Residents. Arch Phys Med Rehabil. 2003;84:1129-1136. doi: 10.1016/S0003-9993(03)00149-7.

14. Yu DT, Platt R, Lanken PN, Black E, et al. Relationship of pulmonary artery catheter use to mortality and resource utilization in patients with severe sepsis. Crit Care Med. 2003;31(12):2734-41. doi: 10.1097/01.CCM.0000098028.68323.64.

15. Moss RR, Humphries KH, Gao M, et al. Outcome of Mitral Valve Repair or Replacement: A Comparison by Propensity Score Analysis. Circulation. 2003;108:90-97. doi: 10.1161/01.cir.0000089182.44963.bb

Changes made: none

Comment 10: 

Page 19, line 248 : might you mean under-estimation?

Response: Thank you for the comment. Line 248 refers to an underestimation of the effect of exposure. The wording of the footnote of the table has been corrected.

Changes made: none

Discussion:

Comment 11:

You state that 40% of Peruvian children between two and three years old would not develop adequate EVC, partly as a result of chronic malnutrition. If malnutrition only explains 8% then it does not seem to be the main source of developmental setbacks in communication? 

Response: Thank you for the observation. Indeed, according to our results, chronic malnutrition only partially explains EVC in the Peruvian population of infants of 24 to 36 month. However, it is known that EVC, similar to other components of ECD, responds to different factors which concomitantly affect the non-development of EVC, such as secure attachment of children and parents, and stimulation of motor development and gesticulation (https://bit.ly/3iXDl3J, https://bit.ly/3dtzsm5, https://bit.ly/2H0q1hT). In addition, chronic malnutrition acts as a cross factor, affecting not only ECV, but also other components of ECD. This characteristic makes chronic malnutrition a cornerstone of the delay of early childhood development and its components, including EVC. Taking this into account and despite the fact that the impact of chronic malnutrition over EVC may seem small, the potential effect and interrelation it could have over other variables magnify this effect.

Changes made: none

Comment 12:

Relatedly, please comment on the extent to which a 8% lower prevalence of EVC is substantial and problematic. This seems like a pretty small effect size. The importance of this effect would also be easier to grasp if you include more information about the outcome variable.

Response: Thank you for the observation. As mentioned in the previous comment, chronic malnutrition explains part of all the components that can influence EVC and early childhood development, and therefore, other variables need to be taken into account, such as whether the child had any hospitalization or the behavior that parents have towards the child. This latter variable could not be measured due to the nature of the secondary data of the study, and could comprehensively explain the impact between the two variables of interest. Likewise, the 8% mentioned can be seen as a small effect, however, this finding can have long-term relevance, potentially in the school or university year and in social development in adults. Therefore, it is necessary to propose strategies that mitigate chronic malnutrition as one of the factors influencing cognitive development with the aim of improving early childhood development and labor productivity in developing countries, like Peru.

Now it reads: “This finding can have long-term relevance, potentially in the school or university year and in social development in adults. Therefore, it is necessary to propose strategies that mitigate chronic malnutrition as one of the factors influencing cognitive development with the aim of improving early childhood development and labor productivity in developing countries, like Peru.”

Comment 13:

You mention some other variables like attachment and socioeconomic factors that may affect EVC, but these would be accounted for by the matching, correct? So there still seems to be an unexplained gap?

Response: Thank you for the observation. It is true that there are different affective variables or factors that can affect EVC, such as secure attachment. However, this was not considered in the study because this variable is measured in ENDES in children up to 12 months of age, while in our study the group of children included were between 24 and 36 months old. This has been included as a limitation of the study.

Now it reads: “It has been mentioned that EVC can be influenced by different affective variables such as secure attachment of children and their parents, and stimulation of motor development and gestures. However, our research included children ranging between 24 and 36 months of age, while the secure attachment variable was collected only in children under 12 months, making it impossible to assess how this variable could affect our results. "

Comment 14:

I agree that the PSM approach is a great strength, but you may discuss some of the limitations of PSM methodologies. King and Nielsen (2016) proposed that PSM has serious limitations. Despite their analysis mostly refers to pair matching without replacement, they made interesting points on the fully blocked randomization approach. This is highly debatable, but still a good way to discuss PSM limitations. In particular given the current trends point toward the use of weighting, something used as a secondary approach.

Response: Thank you for the observation. It is true that the research carried out by King and Nielsen [1] described some of limitations that the application of the PSM method has. According to their findings, this method would often increase the imbalance between the covariates, generating inefficiency, dependence on the model and bias. This topic is included in our study.

1. King G, Nielsen R. Why Propensity Scores Should Not Be Used for Matching. Political Analysis. 2019;27(4):435-454. doi:10.1017/pan.2019.11

Now it reads: “The methodological limitation related to the PSM is based on the application of Common Support (CS) since it tends to eliminate observations in order to achieve a completely random experiment. In this study, the impact of chronic malnutrition in children between 24 and 36 months was estimated, with the application of the PSM by Kernel and the estimate by nearest neighbor, and the estimate of weighted effects with adjusted regression was considered. In addition, a sensitivity analysis was applied to the effect estimates using the Rosenbaum limit approach. These estimates were made with the intention of assessing the impact of the exposure under different methodological perspectives.”

Comment 15:

In Table 1 & 2:

The caption should be complete including the which study population and study site. The questions items should be described in the table and mean/SD should be reported separately. Some questions have been labelled Q1 Q2 that should be described in this table.

Response: Thank you for the observation. We have separated the items in the tables by type of variable (categorical or numerical), to describe the use of mean and standard deviation in numerical variables. Likewise, the titles of Tables 1 and 2 have been changed, and the abbreviations have been specified at the bottom of both tables.

Comment 16:

In Table 3, the caption should include Effective Verbal Communication instead of its abbreviation including the study population, study site and sample size.

Response: Thank you for the observation. We have added the missing information in the title of Table 3 to improve reader understanding.

Now it reads: “Table 3. Exposure effects of stunting on Effective Verbal Communication in 601 Peruvian children aged 24 to 36 months”.

Comment 17:

Table 4: Revise and complete the caption of the table, suggested “sensitivity analysis for estimating the impact of chronic 236 malnutrition on EVC in children aged 24 to 36 months”.

Response: Thank you for the observation. We agree with the title given by the reviewer, therefore, the change from “Sensitivity Analysis” to “Sensitivity Analysis was performed to estimate the impact of stunting on effective verbal communication in 601 Peruvian children aged 24 to 36 months”.

Comment 18:

Study Limitation and its strengths should be mentioned under a spate heading after end of the research paper.

Response: Thank you for the observation. According to the guidelines of the journal, the corresponding changes have been made and the limitations and strengths are now displayed after the discussion.

Comment 19:

The study title should reflect that it is a secondary data analysis so accordingly study title should be revised.

Response: Thank you for the observation. We have changed the title to specify that it is a secondary data study.

Now it reads: “Nutritional status and effective verbal communication in Peruvian children. A secondary analysis of the 2019 Peru Demographic and Health Survey”.

Other changes

To homogenize the term stunting, “chronic malnutrition” was changed to “stunting” for better understanding of the term. Likewise we also homogenize the term effective verbal communication (EVC) along the text.

In accordance with the instructions for authors, we have corrected references to Vancouver style.

King regards, 

The authors

---

## [Decision Letter · Decision Letter 1]

19 Jan 2021

PONE-D-20-26209R1

Nutritional status and effective verbal communication in Peruvian children. A secondary analysis of the 2019 Demographic and Health Survey

PLOS ONE

Dear Dr. Vargas-Fernández,

Thank you for submitting your manuscript to PLOS ONE. After careful consideration, we feel that it has merit but does not fully meet PLOS ONE’s publication criteria as it currently stands. Therefore, we invite you to submit a revised version of the manuscript that addresses the points raised during the review process.

All that remains is including the definition of malnutrition used for this study. Once that's done, it shouldn't need to go back out for review - it can be editorially accepted.

We look forward to receiving your revised manuscript.

Kind regards,

James West, PhD

Academic Editor

PLOS ONE

Reviewers' comments:

Reviewer's Responses to Questions

**Comments to the Author**

1. If the authors have adequately addressed your comments raised in a previous round of review and you feel that this manuscript is now acceptable for publication, you may indicate that here to bypass the “Comments to the Author” section, enter your conflict of interest statement in the “Confidential to Editor” section, and submit your "Accept" recommendation.

Reviewer #1: All comments have been addressed

Reviewer #2: (No Response)

2. Is the manuscript technically sound, and do the data support the conclusions?

Reviewer #1: Partly

Reviewer #2: Yes

3. Has the statistical analysis been performed appropriately and rigorously? 

Reviewer #1: Yes

Reviewer #2: Yes

4. Have the authors made all data underlying the findings in their manuscript fully available?

Reviewer #1: Yes

Reviewer #2: Yes

5. Is the manuscript presented in an intelligible fashion and written in standard English?

Reviewer #1: Yes

Reviewer #2: Yes

6. Review Comments to the Author

Reviewer #1: I have reviewed this paper earlier and now this is the revised version of the paper submitted to PLOS ONE. I recommend to accept the paper for publication.

Reviewer #2: I am mostly satisfied with your response to my comments but I do still think that you should sepefic which definition of malnutrition by WHO you use. Malnutrition is not just stunted growth. https://www.who.int/news-room/fact-sheets/detail/malnutrition.

Otherwise, I think that this is an important and well-executed paper!

7. PLOS authors have the option to publish the peer review history of their article (what does this mean?). If published, this will include your full peer review and any attached files.

Reviewer #1: No

Reviewer #2: No

---

## [Author Response · Author response to Decision Letter 1]

20 Jan 2021

Dr. James West

Academic Editor

PLOS ONE

Title: Nutritional status and effective verbal communication in Peruvian children. A secondary analysis of the 2019 Demographic and Health Survey

Dear Editor and Reviewers,

Thank you for your valuable suggestions and remarks. After incorporating your feedback our manuscript has greatly improved. Please find below the point-by-point response to your comment. We hope that the present version of the manuscript is now acceptable for publication, and we look forward to your feedback.

Comments of Editor:

All that remains is including the definition of malnutrition used for this study. Once that's done, it shouldn't need to go back out for review - it can be editorially accepted. 

Response: Thank you for the observation. For better understanding we have rewritten the sentence.

Now it reads: “The World Health Organization (WHO) classification (defined by variations of more than 2 standard deviations [SD] of the height for age below the 2006 WHO Child Growth Standards median) was used for the exposure variable "stunting" (HW70) [18] and EVC in children aged between 24 to 36 months (R4).”

Other changes

In some cases, “malnutrition” was changed to “stunting” for better understanding of the term.

King regards, 

The authors

---

## [Editor Report · Decision Letter 2]

21 Jan 2021

Nutritional status and effective verbal communication in Peruvian children. A secondary analysis of the 2019 Demographic and Health Survey

PONE-D-20-26209R2

Dear Dr. Vargas-Fernández,

We’re pleased to inform you that your manuscript has been judged scientifically suitable for publication and will be formally accepted for publication once it meets all outstanding technical requirements.

Kind regards,

James West, PhD

Academic Editor

PLOS ONE
---

## [Editor Report · Acceptance letter]

8 Feb 2021

PONE-D-20-26209R2 

Nutritional status and effective verbal communication in Peruvian children. A secondary analysis of the 2019 Demographic and Health Survey 

Dear Dr. Vargas-Fernández:

I'm pleased to inform you that your manuscript has been deemed suitable for publication in PLOS ONE. Congratulations! Your manuscript is now with our production department. 

Kind regards, 

on behalf of

Dr. James West 

Academic Editor

PLOS ONE